# *MITF* Contributes to the Body Color Differentiation of Sea Cucumbers *Apostichopus japonicus* through Expression Differences and Regulation of Downstream Genes

**DOI:** 10.3390/biology12010001

**Published:** 2022-12-20

**Authors:** Lili Xing, Shilin Liu, Libin Zhang, Hongsheng Yang, Lina Sun

**Affiliations:** 1CAS Key Laboratory of Marine Ecology and Environmental Sciences, Institute of Oceanology, Chinese Academy of Sciences, Qingdao 266071, China; 2Laboratory for Marine Ecology and Environmental Science, Qingdao National Laboratory for Marine Science and Technology, Qingdao 266237, China; 3CAS Engineering Laboratory for Marine Ranching, Institute of Oceanology, Chinese Academy of Sciences, Qingdao 266071, China; 4University of Chinese Academy of Sciences, Beijing 100049, China; 5Shandong Province Key Laboratory of Experimental Marine Biology, Qingdao 266071, China; 6The Innovation of Seed Design, Chinese Academy of Sciences, Wuhan 430071, China

**Keywords:** sea cucumber, body color, *MITF*, ChIP-seq, gene expression, gene regulation

## Abstract

**Simple Summary:**

The sea cucumber *Apostichopus japonicus* is an ecologically significant echinoderm with considerable nutritional and therapeutic value. Color variation of *A. japonicus* is one of its most important cultivation traits, as color affects its bioactive compounds, taste, and market price. Microphthalmia-associated transcription factor (*MITF*) is one of the most critical genes in melanocyte development and melanin synthesis pathways. However, how *MITF* regulates body color and differentiation in sea cucumbers is poorly understood. Here, we analyzed the expression level and location of MITF in white, purple, and green sea cucumbers and identified the genes regulated by MITF using chromatin immunoprecipitation followed by sequencing. Our findings suggested that *MITF* contributed to the body color differentiation of green, purple, and white sea cucumbers *A. japonicus* through expression differences and regulation of downstream genes. These findings give researchers a foundation for further research into the processes that underlie the development of body color in sea cucumbers and provide light on how genes are regulated. The results will also provide support for sea cucumber breeding and a useful theoretical basis in order to contribute to the development of the sea cucumber industry.

**Abstract:**

Melanin, which is a pigment produced in melanocytes, is an important contributor to sea cucumber body color. *MITF* is one of the most critical genes in melanocyte development and melanin synthesis pathways. However, how *MITF* regulates body color and differentiation in sea cucumbers is poorly understood. In this study, we analyzed the expression level and location of MITF in white, purple, and green sea cucumbers and identified the genes regulated by MITF using chromatin immunoprecipitation followed by sequencing. The mRNA and protein expression levels of MITF were all highest in purple morphs and lowest in white morphs. In situ hybridization indicated that MITF mRNA were mainly expressed in the epidermis. We also identified 984, 732, and 1191 peaks of MITF binding in green, purple, and white sea cucumbers, which were associated with 727, 557, and 887 genes, respectively. Our findings suggested that *MITF* contributed to the body color differentiation of green, purple, and white sea cucumbers through expression differences and regulation of downstream genes. These results provided a basis for future studies to determine the mechanisms underlying body color formation and provided insights into gene regulation in sea cucumbers.

## 1. Introduction

Sea cucumbers are marine invertebrates that contain numerous bioactive compounds, such as triterpene glycosides, glycosaminoglycans, gangliosides, collagen, branched-chain fatty acid, and lectins, which serve as potential sources of nutraceutical, pharmaceutical, and cosmetic agents [1,2]. Sea cucumbers are popular as a tonic and traditional medicine in China and other Asian countries that are deeply influenced by Chinese traditional culture [3]. In China, 134 species of sea cucumber are recognized. Most species are distributed south of Hainan Island, but the most economically valuable species (*Apostichopus japonicus*) is distributed in northern China [3].

Color variation of *A. japonicus* is one of its most important cultivation traits, as color affects its bioactive compounds, taste, and market price [4,5,6]. The body color of *A. japonicus* may be red, green, black, white, or purple [7,8]. In Japan and South Korea, red, green, and black variants of native *A. japonicus* have been studied [9,10,11], while in China, green *A. japonicus* is the main color morph. Entirely white albino and purple individuals are rarely found in nature, but recently, they were successfully cultivated at the cooperative aquaculture center in Weihai, China [4]. Studies have shown that melanocytes and the melanin they produce play an important role in the formation and differentiation of body color of sea cucumbers [12,13].

Microphthalmia-associated transcription factor (*MITF*), a basic helix–loop–helix leucine zipper factor, governs multiple steps in the development of melanocytes, including specification from the neural crest, growth, survival, and terminal differentiation [14]. Additionally, it activates genes involved in the synthesis of melanin as well as the development and intracellular movement of melanosomes. The melanosome is lysosome-related organelle that is responsible for melanin synthesis and storage [15]. Researchers have demonstrated that the expression level, polymorphisms, and epigenetic modifications of the *MITF* gene could affect the color variation of animals [16,17,18,19].

Identifying the genomic binding sites of transcription factors (TFs) is important for determining the transcriptional regulatory networks involved in biological processes. The combination of chromatin immunoprecipitation and second-generation sequencing technology (ChIP-seq) can efficiently screen potential DNA binding sites that interact with TFs throughout the genome. It has become a mainstream method in genomics and epigenomics and has led to important discoveries related to disease-associated transcriptional regulation [20,21,22], tissue specificity of epigenetic regulation [23], and chromatin organization [24,25]. ChIP-seq profiles recently discovered a large number of candidate MITF targets across different categories, including DNA replication and repair, cell proliferation, and mitosis [26,27,28]. The wide variety of MITF-target gene functions supports the significance of this TF in melanocyte formation and color variation [14].

Zhao et al. (2012) cloned the full-length MITF cDNA from *A. japonicus* and reported that it was 3104 base pairs (bp) long, with an open reading frame that starts at 587 bp and ends at 2083 bp, totaling 1497 bp and encoding 499 amino acids. The basic helix–loop–helix leucine structure starts at amino acid 279 and ends at amino acid 340 [29]. However, how MITF regulates the body color formation and differentiation of sea cucumbers remains to be resolved. Here, we evaluated the expression levels of MITF mRNA and protein in green, white, and purple sea cucumbers and revealed the distribution of MITF expression in the body wall of sea cucumbers. What is more, ChIP-seq assay were employed to analyze genes regulated by MITF in three color morphs of sea cucumbers. These data improved our understanding of key sequence motifs and regulatory relationships in sea cucumbers. Our findings also shed light on gene function and the regulatory mechanisms behind the body color formation and differentiation.

## 2. Materials and Methods

### 2.1. Ethics Statement

No specific permission was required for the collection of sea cucumbers, and all experiments were conducted with the approval of the Experimental Animal Ethics Committee, Institute of Oceanology, Chinese Academy of Sciences, Qingdao, Shandong Province, China.

### 2.2. Sample Collection

The green *A. japonicus* (G), white *A. japonicus* (W), and purple *A. japonicus* (P) used in this experiment were collected from Shandong Oriental Ocean Sci-Tech Co. Ltd. (Yantai City, Shandong Province, China). The animals were transferred to the laboratory and maintained in natural seawater (water temperature, 14 °C; salinity, 30%) for 2 days. All sea cucumbers used in this study weighed about 50 g. Fifty individuals of each color morph were selected after acclimatization. The body wall was sampled and immediately frozen in liquid nitrogen.

### 2.3. Real-Time Polymerase Chain Reaction (PCR) Analysis

For each color morph, three replicates of RNA samples, each of which contained tissue from three individuals, were analyzed. The total RNA of each sample was isolated from the sea cucumber body wall using RNeasy Kit (Qiagen, TX, USA). Primer3 (v0.4.0; http://bioinfo.ut.ee/primer3-0.4.0/primer3/; 30 August 2018) was used to design MITF primers. The forward primer of MITF was 5′-ACATGGGACACACCGCACAGGTTACGGC-3′, and the reverse primer was 5′-CCTCATTGATGCCCCCTCCTAAAGACC-3′. β-actin was used as a reference gene for internal standardization. The forward primer of β-actin was 5′-CATTCAACCCTAAAGCCAACA-3′, and the reverse primer was 5′-TGGCGTGAGGAAGAGCAT-3′. First-strand cDNA was synthesized using reverse transcriptase (Takara Bio Inc., Kusatsu, Japan). The expression levels of MITF mRNA were determined using the SYBR Green real-time PCR assay with an Eppendorf Mastercycler ep realplex (Eppendorf, Hamburg, Germany). Additional detailed experimental information was the same as previously described [13]. The 2^−ΔΔCT^ method was used to calculate expression levels.

### 2.4. Western Blot Analysis

To determine the protein expression level of MITF, a Western blot experiment was performed in green *A. japonicus* (n = 3), white *A. japonicus* (n = 3), and purple *A. japonicus* (n = 3). The internal reference protein in this experiment was β-actin. Body wall tissue (30 mg) from each experimental sample was ground in liquid nitrogen, and then 200 μL of radioimmunoprecipitation assay buffer containing 1 mMol/L phenylmethanesulfonyl fluoride was added. Each sample was homogenized three times at low temperature and high speed for 10 s each time and then lysed for 2 h on ice. The samples were centrifuged (12,000× *g*) at 4 °C, and the supernatant was retained for bicinchoninic acid (BCA) protein quantification. For this assay, 200 μL of BCA solution was added to the wells of a microplate and mixed with 4 μL of copper sulfate solution to prepare the quantitative working solution. The solution was incubated at 37 °C for 30 min.

Next, a polyacrylamide gel was prepared with a 5% concentrated gel and a 6% separating gel, with a loading volume of 30 μg/well. Subsequently, the concentrated gel was electrophoresed at 80 V for 30 min and the separation gel at 120 V, and the electrophoresis stop time was determined by pre-staining the protein marker. The PVDF membrane was wet transferred at 280 mA for 2.5 h, then completely submerged in 5% bovine serum albumin–Tris-buffered saline with Tween (TBST) with gentle shaking at room temperature for 60 min. The primary antibody was MITF antibody. We obtained the protein of sea cucumber MITF (GenBank: HQ606465.1) that was previously expressed in vitro. The plasmid vector used was pet28a. The full-length sequence expressed protein of sea cucumber MITF was used as an immunogen. Then, we commissioned Sangon Biotech Co., Ltd. (Shanghai, China), to stimulate rabbits and to produce MITF polyclonal antibody. This MITF antibody was specific to sea cucumber. The full-length sequence expressed protein of sea cucumber MITF was used as immunogen to stimulate rabbits, and polyclonal antibody was obtained. This MITF antibody was specific to sea cucumber. The antibody was diluted with 5% skim milk, added to the membrane, incubated at room temperature for 30 min, and kept at 4 °C overnight. The membrane was washed five times with TBST for 3 min each time. To identify the corresponding primary antibody, the horseradish-peroxidase-labeled secondary antibody (Goat Anti-Rabbit IgG (H + L) HRP, D110058-0100, BBI, 1:8000) was diluted with 5% skim milk and added to the membrane, which was shaken gently for 1 h at room temperature. The membrane was washed six times with TBST for 3 min each time. Next, enhanced chemiluminescence was used to analyze the membrane. In the darkroom, the membrane was covered with an X-OMAT BT film (Sangon, Shanghai, China). Subsequently, the film was placed in an X-ray film processor for developing and fixing, and the protein bands were obtained. NIH Image 1.63 software (Syngene, Cambridge, UK) was used to quantitatively measure each protein band on the film to obtain the relative expression of the target protein.

### 2.5. In Situ Hybridization Analysis

The reagents used in this experiment were as follows: (1) 20× SSC: 175.3 g/L sodium chloride, 88.2 g/L sodium citrate, with pH adjusted to 7.0; (2) denaturation solution: formamide 70%, 2× SSC, 0.1 mM ethylenediaminetetraacetic acid (EDTA); (3) hybridization buffer: 300 μL of formamide, 12 μL of 50× Denhardt’s solution, 120 μL of 50% dextran sulfate, 9 μL of 10 μg/μL yeast tRNA, 49 μL of RNase-free dH2O; (4) proteinase K reaction solution: 100 mM Tris-HCL, 50 mM EDTA, 1 μg/mL proteinase K; (5) wash solution- I (WS-I): formamide 50% (*V*/*V*), 20× SSC 10% (*V*/*V*), diethyl pyrocarbonate (DEPC) water 40% (*V*/*V*); (6) WS-II: 4× SSC, 0.05% Tween-20; (7) WS-III: 4 SSC. All the above reagents were treated with 0.1% DEPC water and sterilized overnight before use.

The MITF probe was designed with the probe sequence 5′-FAM-CUGUCAGGGUAAUCGGUUCCUGCUUGAUGGUCG-3′. The probe sequence was synthesized artificially, and the FAM marker was added to the 5′ during the synthesis. The type of this probe was an oligonucleotide probe with higher hybridization efficiency. The body wall samples of white, green, and purple *A. japonicus* were stored in paraformaldehyde, and the samples without probes were used as negative controls. Paraffin-embedded specimens were sectioned at a thickness of 5 μm, and sections were placed on slides, which were then baked in a 60 °C oven for 30 min. Subsequently, the sections were dewaxed three times with xylene in a fume hood for 5 min each time and rinsed with absolute ethanol twice for 2 min each time. The slides were placed on a rack, immersed in proteinase K reaction solution, placed in a 37 °C water bath, and incubated for 20 min. Next, the slides were rinsed with 2× SSC three times for 2 min each time at room temperature. They were subjected to gradient ethanol dehydration (−20 °C precooling) at 50%, 70%, 90%, and 100%. The slides were placed in a staining tank containing denaturing liquid for 8 min at 75 °C, and then the hybrid solution containing DNA probes was added, followed by incubation at 75 °C for 8 min. The slides were moved into the dye cylinder containing pre-cooled 70% alcohol for 2 min and then into pre-cooled 80%, 90%, and 100% alcohol for 2 min each time, and then they were allowed to air dry. Finally, 50 μL of denatured probes were added to the section samples, which were placed in a wet box. The samples were hybridized in an oven at 42 °C overnight (12–16 h) and then eluted in WS-I, WS-II, and WS-III, respectively. The section samples were observed and photographed using a fluorescence microscope (Olympus IX71, Tokyo, Japan).

### 2.6. Chromatin Immunoprecipitation Sample Preparation

ChIP was performed as previously described with minor modifications [30]. First, 1 g of sea cucumber body wall tissue was cross-linked with 1% formaldehyde for 10 min at 37 °C with constant shaking. Then, the addition of 125 mM glycine prevented cross-linking. Chromatins were collected on ice after samples were lysed. The collected chromatins were sonicated by Bioruptor (Diagenode, Liège, Belgium) at standard mode for 30 sec to obtain soluble sheared chromatin (average DNA length of 200–500 bp). Next, 100 μL of chromatin was utilized for immunoprecipitation using MITF antibodies, and 20 μL of chromatin was maintained at −20 °C for input DNA. The immunoprecipitation procedure was conducted at 4 °C for an overnight period using 10 micrograms of antibody. Then, 30 μL of protein beads was added the following day, and the samples were then incubated for an additional 3 h. After that, the beads were washed with different solutions [29]. RNase A (final concentration 8 g/mL) was applied first for 6 h at 65 °C, followed by proteinase K (final concentration 345 g/mL) for an overnight incubation at 45 °C. Immunoprecipitated DNA was used to construct sequencing libraries.

### 2.7. Illumina Sequencing

ChIP-Seq libraries were prepared and sequenced following ENCODE guidelines [31] by Wuhan IGENEBOOK Biotechnology Co., Ltd. (Wuhan, China) [32]. Following repair and adaptor ligation procedures, DNA fragments (250–350 bp) were chosen using SPRI beads and amplified by PCR for 15 cycles. The Bioanalyzer 2100 (Agilent, Santa Clara, CA, USA) and Qubit fluorometer (Invitro-gen, Carlsbad, CA, USA) were used to validate libraries. The ChIP-seq libraries were sequenced using Illumina X ten with the PE 150 method (San Diego, CA, USA). Low-quality readings were eliminated using Trimmomatic (version 0.38). Clean reads were mapped to the *A. japonicus* genome using Bwa (version 0.7.15). Potential PCR duplicates were eliminated using Samtools (version 1.3.1). MACS2 software (version 2.1.1.20160309) was used to identify the high-confidence read enrichment regions (peaks) based on default parameters (bandwidth, 300 bp; model fold, 5, 50; q value, 0.05). If the summit of a peak was located closest to the transcription start site of one gene, the peak was assigned to that gene [33]. HOMER (version3) was used to predict motif occurrence within peaks, with default settings for a maximum motif length of 12 bp. The EasyGO gene ontology enrichment analysis tool (http://bioinformatics.cau.edu.cn/easygo/; 10 May 2022) was used to perform the gene ontology (GO) enrichment analysis. GO term enrichment was calculated using the hypergeometric distribution with a P value cutoff of 0.01. In order to identify overrepresented GO terms, false discovery rate for multiple comparisons was used to correct P values from Fisher’s exact test. Kyoto Encyclopedia of Genes and Genomes (KEGG) enrichment analysis was carried out using clusterProfiler (http://www.bioconductor.org/packages/release/bi-oc/htmL/clusterProfiler.htmL; 10 May 2022) in the R package to uncover probable roles of genes.

### 2.8. Statistical Analysis

Statistical analysis was performed using SPSS 18 software (SPSS, Inc., Chicago, IL, USA). The data of mRNA and protein expression level were presented as the mean ± standard deviation (n = 3), and they were statistically analyzed by one-way ANOVA with a Tukey test. *p* values <0.05 were considered statistically significant. In addition, correlation testing between mRNA and protein expressions was also conducted.

## 3. Results

### 3.1. Difference in MITF Gene Expression among the Three Color Morphs of Sea Cucumbers

The relative abundances of MITF mRNAs in the body wall of white, green, and purple sea cucumbers were 0.18, 0.53, and 0.91, respectively. The expression level of MITF RNA was highest in the purple morph and lowest in the white morph (*p* < 0.05) (Figure 1A). In green sea cucumbers, it was more than twice as high as that in white sea cucumbers. The mRNA expression of MITF in purple sea cucumber was as much as four times higher than that in white sea cucumber. Western blot results showed that the expression levels of MITF protein followed the same pattern as that of mRNA among the three color morphs (purple > green > white) (Figure 1B). The relative protein expressions of MITF in white, green, and purple sea cucumbers were 0.14, 0.22, and 0.25, respectively. The expression of MITF protein in white sea cucumbers was significantly lower than that in green and purple sea cucumbers (*p* < 0.05). The correlation test between mRNA and protein expressions showed that they were significantly correlated (*p* < 0.01).

### 3.2. Spatial Expression of MITF at the mRNA Levels

The spatial expressions of MITF mRNA were visualized using in situ hybridization (Figure 2). In Figure 2, panels W1, G1, and P1 demonstrate the location of MITF mRNA in sea cucumbers; panels W2, G2, and P2 were negative controls without fluorescent probes of MITF, which indicated the tissue structure of the sea cucumber body wall according to the DAPI nuclear dye. The cuticle, epidermis, and dermis tissue could be distinguished in the body wall of sea cucumbers. A previous study showed that the epidermal layer was thinnest in white sea cucumbers, followed by green and then purple morphs, and melanocytes were primarily distributed in the epidermal layer [13]. In this study, we found no difference in the location of MITF expression in white, green, and purple sea cucumbers, which were all expressed mainly in the epidermis.

### 3.3. ChIP-Seq Data Analysis

In green, purple, and white sea cucumbers, the MITF chip samples were designated IP_G, IP_P, and IP_W, respectively, whereas the corresponding MITF input samples were designated In_G, In_P, and In_W. Following completion of the sequencing, the raw data were processed by de-adapting and deleting low-quality data to provide useful data. Finally, 48, 38, 41, 56, 43, and 55 million clean reads were obtained for IP_G, IP_P, IP_W, In_G, In_P, and In_W samples, respectively (Table 1). After quality control, the clean reads were mapped to the *A. japonicus* genome [34]. The mapping rates of the six samples were 50.62%, 69.24%, 48.95%, 82.34%, 82.88%, and 82.36%, respectively (Table 2). The green, purple, and white sea cucumber samples yielded 984, 732, and 1191 peaks, respectively, with average lengths of 355, 386, and 346 bp (Table 3).

### 3.4. Annotation of Genes Identified by MITF ChIP

The peaks identified in this study were associated with 727, 557, and 887 genes in green, purple, and white sea cucumbers, respectively. Notably, more than 60% of these MITF binding sequences were located distal to the transcription start sites of their target genes (Figure 3A–C). Appendix A show the specific distribution information of peaks in green, purple, and white sea cucumbers, respectively. Significantly enriched KEGG pathways of genes identified by MITF ChIP in green sea cucumbers included tyrosine metabolism; tropane, piperidine, and pyridine alkaloid biosynthesis; ribosome; protein processing in endoplasmic reticulum; protein kinases; and prokaryotic defense system (Figure 3D). In purple sea cucumbers, significantly enriched KEGG pathways were tropane, piperidine, and pyridine alkaloid biosynthesis; thiamine metabolism; sulfur metabolism; RNA polymerase; and phenylalanine, tyrosine, and tryptophan biosynthesis (Figure 3E), and in white sea cucumbers they were the Wnt signaling pathway; sphingolipid signaling pathway; riboflavin metabolism; retinal metabolism; and protein kinases (Figure 3F).

### 3.5. Comparison of Genes Identified by MITF ChIP in Different Color Morphs of Sea Cucumbers

Appendix A shows the MITF binding sequences that were common among the three color morphs of *A. japonicus*. Common MITF binding genes included trichohyalin, 60S ribosomal protein L6, tyrosine kinase receptor Cad96Ca, calcium/calmodulin-dependent protein kinase type IV, acetyl-coenzyme A transporter 1, glycoprotein 3-alpha-L-fucosyltransferase A, alanine-tRNA ligase, GMP reductase 2, and GTPase IMAP family member 4. To identify differences in the genes regulated by MITF among different sea cucumber color morphs, we analyzed the genes identified by MITF ChIP in purple and white sea cucumbers in comparison with those identified in green sea cucumbers. Appendix A show the genes differentially bound to MITF in purple and white sea cucumbers, respectively, compared with those in green sea cucumbers. Subsequently, we performed GO and KEGG enrichment analysis of these genes (Figure 4). Compared with green sea cucumbers, the differentially binding genes identified by MITF ChIP in purple sea cucumbers were significantly enriched in GO terms related to negative regulation of megakaryocyte differentiation, telomere organization, CENP-A containing nucleosome assembly, replication-dependent nucleosome assembly, and RNA-directed DNA polymerase activity (Figure 4A), and related KEGG pathways included thiamine metabolism, cysteine and methionine metabolism, lipoic acid metabolism, apelin signaling pathway, and p53 signaling pathway (Figure 4B). Compared with green sea cucumbers, the differentially binding genes identified by MITF ChIP in white sea cucumbers were significantly enriched in GO terms related to negative regulation of gene expression, epigenetic, negative regulation of megakaryocyte differentiation, chromatin silencing at rDNA, regulation of megakaryocyte differentiation, telomere organization, and nuclear nucleosome (Figure 4C), and related KEGG pathways included nicotinate and nicotinamide metabolism, MAPK signaling pathway, riboflavin metabolism, hippo signaling pathway, cAMP signaling pathway, and ErbB signaling pathway (Figure 4D).

### 3.6. Motif Analysis

Compared with green sea cucumbers, 18 de novo motifs of MITF-specific binding peaks in purple sea cucumbers were enriched. When DNA sequences corresponding to each peak were compared with existing TF binding site motifs to obtain all possible TF binding sites, 11 motifs were found to be enriched. In white sea cucumbers, 20 de novo motifs and 22 known motifs of MITF-specific binding peaks were enriched compared with green sea cucumbers. Table 4, Table 5, Appendix A show the results of known motif enrichment of MITF-specific binding peaks in purple and white sea cucumbers, respectively, compared with green sea cucumbers. The IRF motif was the top identifier for MITF-specific binding peaks in purple sea cucumbers, and other known motifs included Nkx3.1, LXRE, RORgt, CRE, NFkB-p65, RAR: RXR, PR, HOXD13, PAX5, and FOXA1: AR (Table 4 and Appendix A). HRE, THRa, and Stat3 motifs were the top three identifiers for MITF-specific binding peaks in white sea cucumbers, and other known motifs included GEI-11, Foxh1, Elk4, FHY3, Tbx20, LXRE, Tcf3, Mef2b, Tcf4, RAR: RXR, and NF1 (Table 5 and Appendix A).

## 4. Discussion

MITF is the earliest known marker of commitment to the melanocyte lineage, and most of the signaling molecules or TFs implicated genetically in melanocyte development affect either MITF expression or its function [35]. B éjar et al. (2003) proposed a role for MITF as a master developmental regulator, the expression of which is sufficient to initiate and complete the differentiation of pluripotent cells into melanocytes in medaka (*Oryzias latipes*). In sea cucumbers, melanocytes are primarily distributed in the epidermal layer. We found that the thinner epidermal layer of white *A. japonicus* contained fewer epidermal melanocytes than the thicker layers of green and purple morphs. Xing et al. (2018) previously reported that purple sea cucumbers had the thickest epidermal layer of the body wall and the most melanocytes. In the current study, we also found that MITF was expressed in the body wall in all three color morphs but was mainly concentrated in the epidermis. The mRNA and protein expression levels MITF were all highest in purple morphs and lowest in white morphs. Based on these findings, we propose that MITF expression is closely correlated with melanocyte differentiation in sea cucumbers and that the differences in melanocyte density among the three color types of sea cucumbers result from differences in MITF expression. In addition, melanin plays a key role in the formation of sea cucumber body color. The melanin content varies greatly among the three color morphs of *A. japonicus*, and Xing et al. (2017) reported that the content in purple sea cucumbers was more than twice that of green sea cucumbers and almost no melanin was detected in white sea cucumbers. Our results suggested that MITF played a crucial role in melanin synthesis and pigmentation in a gene-dosage-dependent pattern and that it may be involved in body color differentiation.

We identified the binding sites of MITF and characterized its target genes in green, purple, and white sea cucumbers. According to the gene annotation results, the genes regulated by MITF in *A. japonicus* were involved in various functions, including signal transduction, lipid and amino acid metabolism, retinol metabolism, posttranslational modification, exosome and defense, and protein and enzyme synthesis. Thus, MITF played a variety of important roles in sea cucumber immunity and growth in addition to affecting melanin synthesis. The Notch signaling pathway is an essential cell–cell interaction mechanism that regulates processes such as cell proliferation, cell fate decisions, differentiation, and stem cell maintenance [36]. Golan et al. (2015) found that Notch signaling altered the DNA binding capacity of MITF, thus inhibiting the transcriptional program that it mediates. Golan and Levy (2019) further demonstrated that, in addition to the known competition-based repression mechanism of MITF transcriptional activity, Notch signaling activation inhibited the transcription of MITF, leading to a decrease in MITF expression. In the pigmentary system, members of the Notch signaling pathway are expressed in melanocytes [37] and seem to be upregulated in melanoma cell lines [38]. Yue et al. (2015) proposed that the Notch signaling pathway plays a crucial role in clam (*Meretrix meretrix*) shell pigmentation in a gene-dosage-dependent pattern and also is potentially involved in shell color patterning. They also proposed that the calcium signaling pathway may be equally involved in shell color formation via activation of this pathway [39]. In the current study, the genes bound by MITF in green sea cucumbers were significantly enriched in the Notch and calcium signaling pathways, indicating that MITF acted synergistically with these signaling pathways to maintain homeostasis of melanocytes and melanin formation in green sea cucumbers.

Hippo signaling pathway components are structurally and functionally conserved. This pathway has diverse functions, including development, tissue homeostasis, wound healing and regeneration, immunity, and tumorigenesis [40]. In this study, the genes bound by MITF in all three color morphs of *A. japonicus* were significantly enriched in the hippo signaling pathway and also in the exosome in purple sea cucumbers. Exosomes are extracellular vesicles that contain constituents (protein, DNA, and RNA) of the cells that secrete them, and they are associated with immune responses [41]. We hypothesize that purple sea cucumbers possess better immunity through MITF and its regulated hippo signaling pathway and exosome-related genes compared to the other two color morphs. In white sea cucumbers, the genes bound by MITF were enriched in multiple signaling pathways, including Wnt, MAPK, ErbB, cAMP, and apelin signaling pathways. Signaling pathways ultimately exert their influence on cell behavior by regulating the activity of TFs that drive gene expression programs associated with specific cell phenotypes. Recently, Ngeow et al. (2018) showed that GSK3, which is downstream from both the PI3K and Wnt pathways, and BRAF/MAPK signaling converged to control MITF nuclear export. Although MITF expression was lowest in white sea cucumbers compared to the purple and green morphs, we found that it bound the largest number of genes and had diverse functions in this morph, making it a good target for studying gene regulation in the future.

## 5. Conclusions

In conclusion, we evaluated the expression levels and distribution of MITF in different color morphs of *A. japonicus*, and we used a ChIP-seq assay to analyze genes regulated by MITF in green, white, and purple sea cucumbers. The binding sites identified by ChIP-seq improved our understanding of key sequence motifs and regulatory relationships in sea cucumbers. Our results also provided insights into gene function and the mechanisms of body color formation and differentiation.

## Figures and Tables

**Figure 1 biology-12-00001-f001:**
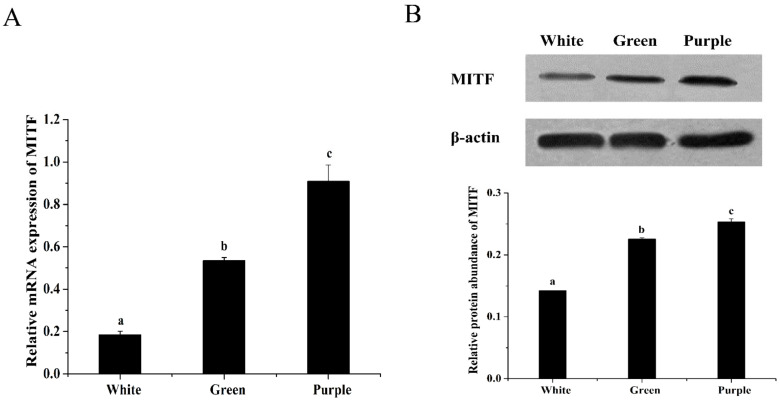
Relative expression levels of MITF mRNA and protein in the body wall of white, green, and purple *A. japonicus*. (**A**) The MITF mRNA expression levels relative to β-actin mRNA levels were examined by qPCR, and the difference among different color morphs was analyzed. (**B**) Total protein (10 μg) was loaded in each line for Western blot analysis, and *β*-actin was used as a loading control. The relative protein abundance of MITF was calculated as relative expression values to the control group. Different letters (a, b, c) indicate significant differences among different color morphs (*p* < 0.05).

**Figure 2 biology-12-00001-f002:**
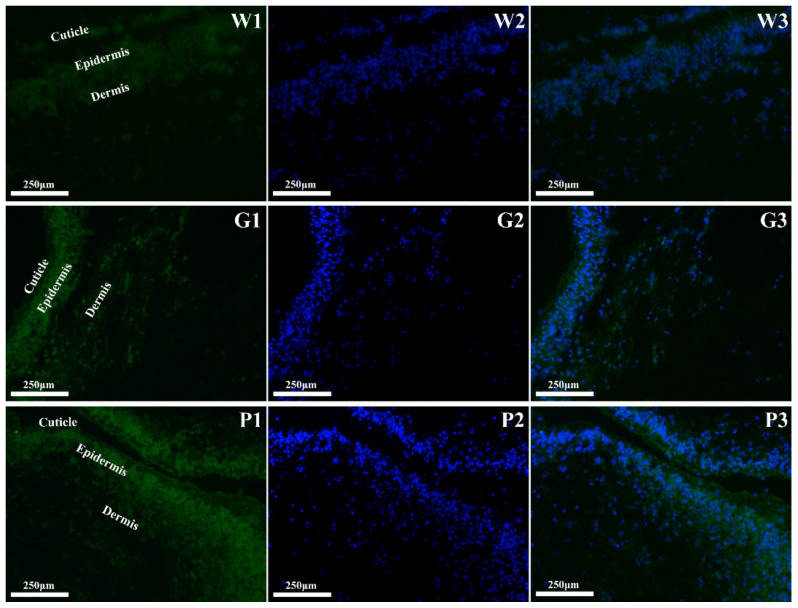
Fluorescence in situ hybridization analysis for MITF in the body wall of white, green, and purple *A. japonicus*. W1: hybridization signals of MITF in white *A. japonicus*; W2: negative control with nuclear dye DAPI in white *A. japonicus*; W3: merged signals in white *A. japonicus*; G1: hybridization signals of MITF in green *A. japonicus*; G2: negative control with nuclear dye DAPI in green *A. japonicus*; G3: merged signals in green *A. japonicus*; P1: hybridization signals of MITF in purple *A. japonicus*; P2: negative control with nuclear dye DAPI in purple *A. japonicus*; P3: merged signals in purple *A. japonicus*.

**Figure 3 biology-12-00001-f003:**
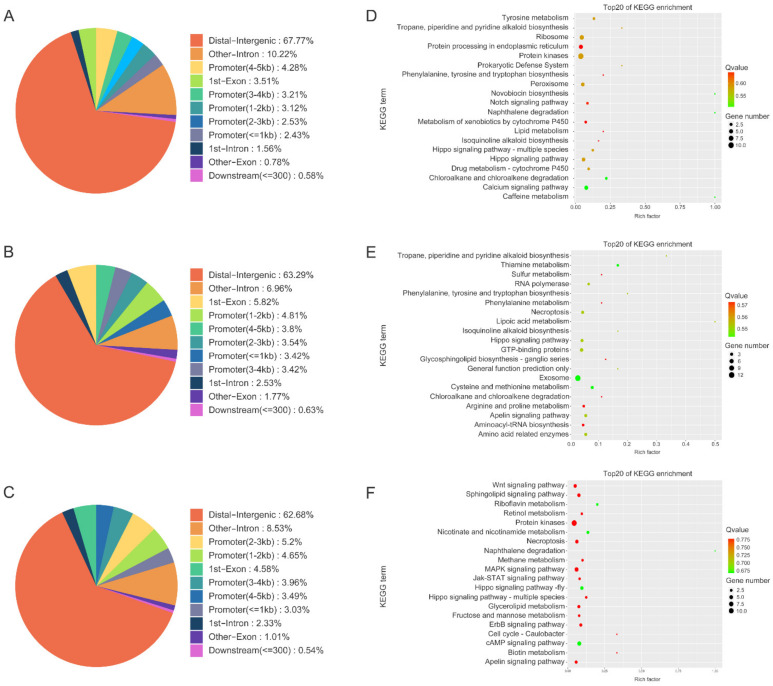
Peak analysis of MITF binding sequence in green, purple, and white *A. japonicus*. (**A**) Peak distribution across genome of green *A. japonicus*. (**B**) Peak distribution across genome of purple *A. japonicus*. (**C**) Peak distribution across genome of white *A. japonicus*. (**D**) KEGG pathway analysis of genes regulated by MITF in green *A. japonicus*. (**E**) KEGG pathway analysis of genes regulated by MITF in purple *A. japonicus*. (**F**) KEGG pathway analysis of genes regulated by MITF in white *A. japonicus*.

**Figure 4 biology-12-00001-f004:**
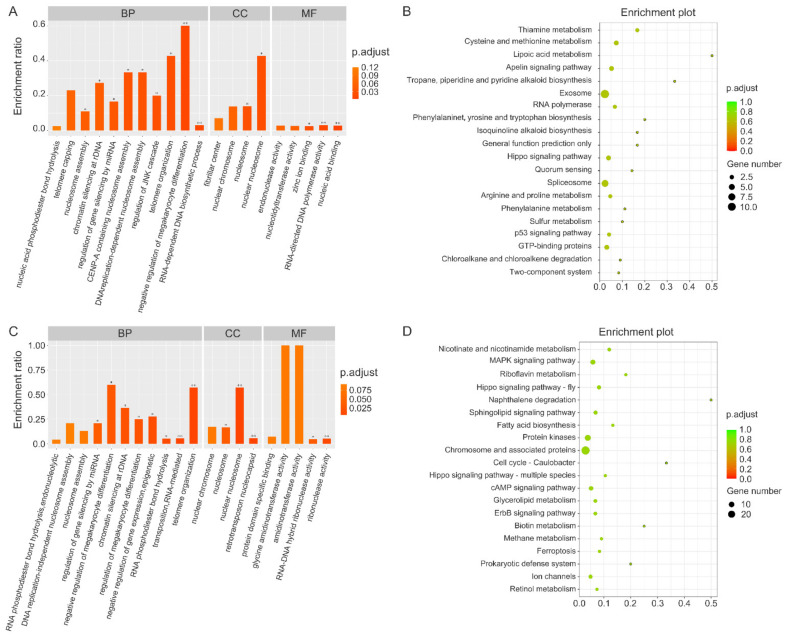
GO and KEGG analysis of genes differentially bound to MITF in purple and white sea cucumber, compared with those in green sea cucumber. (**A**) Gene Ontology classification in purple *A. japonicus*. (**B**) KEGG pathway analysis in purple *A. japonicus*. (**C**) Gene Ontology classification in white *A. japonicus*. (**D**) KEGG pathway analysis in white *A. japonicus*. “*” indicates significant differences with *p* < 0.05; “**” indicates significant differences with *p* < 0.01.

**Table 1 biology-12-00001-t001:** Statistical summary of ChIP-seq data output.

Sample	Raw Reads	Clean Reads	Bases	Q20	Q30	Error Rate	GC Content
IP_G	50,834,380	48,211,710	6,965,268,667	98.32%	95.18%	0.01%	44.48%
IP_P	40,186,312	38,936,494	5,602,175,868	98.47%	95.86%	0.01%	39.89%
IP_W	42,857,474	41,156,574	6,008,796,738	98.55%	95.87%	0.01%	45.03%
In_G	58,644,000	56,691,046	8,203,781,133	98.18%	94.98%	0.01%	37.86%
In_P	44,763,546	43,446,150	6,325,150,652	98.49%	95.92%	0.01%	37.81%
In_W	58,268,370	55,632,032	8,109,868,722	98.51%	95.96%	0.01%	37.62%

**Table 2 biology-12-00001-t002:** Genome mapping of clean reads.

Sample	Total Reads	Mapped Reads	Map Rate	Paired	Single
IP_G	48,321,829	24,458,808	50.62%	22,989,908	706,691
IP_P	39,062,377	27,048,644	69.24%	25,214,094	799,505
IP_W	41,246,498	20,189,447	48.95%	19,071,040	543,201
In_G	56,932,965	46,878,697	82.34%	43,263,590	1,590,244
In_P	43,646,150	36,173,208	82.88%	33,337,656	1,173,866
In_W	55,946,442	46,076,330	82.36%	42,772,052	1,344,408

**Table 3 biology-12-00001-t003:** Peak information statistics.

Sample	Peak Number	Peak Total Length	Peak Average Length	Average Tag
IP_G	984	349,932	355	922
IP_P	732	282,705	386	974
IP_W	1191	412,510	346	706

**Table 4 biology-12-00001-t004:** HOMER enrichment results for top 5 known motifs in purple *A. japonicus*, compared with green *A. japonicus*.

Rank	Motif	Log (*p* Value)	Name
1	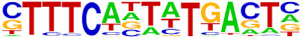	−11.73	IRF:BATF(IRF:bZIP)/pDC-Irf8-ChIP-Seq(GSE66899)/Homer
2	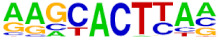	−9.47	Nkx3.1(Homeobox)/LNCaP-Nkx3.1-ChIP-Seq(GSE28264)/Homer
3	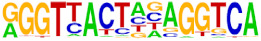	−9.08	LXRE(NR),DR4/RAW-LXRb.biotin-ChIP-Seq(GSE21512)/Homer
4	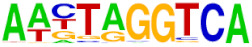	−7.66	RORgt(NR)/EL4-RORgt.Flag-ChIP-Seq(GSE56019)/Homer
5	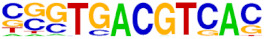	−7.28	CRE(bZIP)/Promoter/Homer

**Table 5 biology-12-00001-t005:** HOMER enrichment results for top 5 known motifs in white *A. japonicus*, compared with green *A. japonicus*.

Rank	Motif	Log (*p* Value)	Name
1	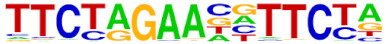	−13.47	HRE(HSF)/Striatum-HSF1-ChIP-Seq(GSE38000)/Homer
2	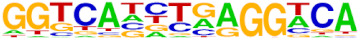	−10.92	THRa(NR)/C17.2-THRa-ChIP-Seq(GSE38347)/Homer
3	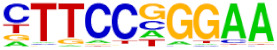	−10.24	Stat3(Stat)/mES-Stat3-ChIP-Seq(GSE11431)/Homer
4	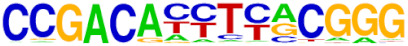	−9.73	GEI-11(Myb?)/cElegans-L4-GEI11-ChIP-Seq(modEncode)/Homer
5	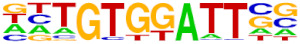	−8.97	Foxh1(Forkhead)/hESC-FOXH1-ChIP-Seq(GSE29422)/Homer

## Data Availability

Not applicable.

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
