# Peer review of "MITF Contributes to the Body Color Differentiation of Sea Cucumbers Apostichopus japonicus through Expression Differences and Regulation of Downstream Genes"

_biology, 2022, doi:10.3390/biology12010001_

Round 1

Reviewer 1 Report

In this manuscript, Xing et al. performed an interesting analysis of the sea cucumber body colour differentiation with an emphasis on the critical transcription factor MITF and its potential downstream regulating genes. The authors have compared the gene and protein expression level of this gene in three color types of sea cucumber, localize its expression and a CHIP assay on the potential downstream genes. In general, this work is radical and would broaden our understanding of body color differentiation in marine wild-type invertebrates.

However, based on the results presented, it may be a bit difficult to find the data that underpins the authors’ conclusions. I raised a number of concerns here.

1)     There seem at least three different color morphotypes in sea cucumber, i.e. purple, green and the albino. What is the concentration of melanin in these three types? I am particularly curious about the green type, is it due to a mixture of variable pigments? Would other pigments also be produced by melanocytes?

2)     In the CHIP analysis, the green morphotype was treated as the “control” group. Without a former explicit introduction as mentioned in my last concern, I found it quite inappropriate to do so. If the melanin concentration decreases in the order of purple, green and white ones, the white one shall be served as the control group? Also, from the technical point of view, it is unclear how the authors perform the statistics?

3)     This may be the most critical concern. Since MITF is critical in the melanin synthesis pathways, what is the role it plays in the albino morphotype, where melanin is supposed to be very low?

Several minor points:

Overall for the figures, first of all, I encourage the authors to include the statistics in the figures. Wherever statistics have been derived (e.g. error bars, statistical significance), the legend needs to provide (Fig. 1 and 2). Also, please ensure that the figures are accessible to colour-blind readers, I encourage the authors to use alternative colour schemes. Finally, the colour code used in Figure 5 is way too sharp.

The title is a bit long, can just say ”MITF contributes to the body colour differentiation of sea cucumbers Apostichopus japonicus through expression differences and regulation of downstream genes”?

Line 232, shall be “Illumina X Ten”.

Line 236, better to briefly explain what ‘peak’ is, as this could be multiple means. 

Figures 3 & 4, quite confusing and redundant. I assume the expression of MITF shall be the same in all three types of sea cucumbers at both the mRNA level and protein level. In this regard, it is not necessary to show all of the results, because they look the same. Additionally, in figure 4, it is not clear the location of MITF, because it seems that the expression is everywhere in the tissue not limited to the arrow-pointing spots.

Line 303, please add a citation of the A. japonicus genome.

Reviewer 2 Report

The manuscript presented by Xing et al. investigated the genetic basis of the body colors of sea cucumbers. The authors profiled and localized MITF expressions at mRNA and protein levels. A set of ChIP-seq peaks were identified in subsequent experiments. Here I have some concerns:

1.    A report for quality control of ChIP-seq experiments is needed, such as JSD value (please take a look at ENCODE’s recommendation).

2.    Line 319, “MITF acts primarily on enhancer regions”, this inference is too arbitrary. If this inference is right, only linking  ChIP-seq peaks to closet genes is inappropriate. This is because enhancers are usually far away from target genes. A better for target gene identification is needed.

3. whether all peaks located on the same chromosome where the MITF gene is (i.e. cis/trans regulation)? Peak distribution on chromosomes should be mentioned?

3.    Lines such as 348, 354, and 355, the use of “differentially expressed genes” is inappropriate

4.    Scientific writing needs to be improved thoroughly. For instance, section 3.2 only has two lines (274-275) expressing results (What you observed should be clearly described), and the rest is background/discussion. Another example is lines 296-307, in which much content can go to methods or figure legends.

5.    Figures 1 and 2 actually can be merged into one figure with two panels. A correlation test between mRNA and protein expressions should be added.

6.    Tables 1 and 2 only summary statistics, thus they might be put in supplemental materials. Instead, part of Table S1 (probably genes with functional annotation) can be made as the main Table.

7.    The melanin synthesis pathway is well studied, please take a look at KEGG database. Therefore, how many genes you identified overlapped with genes in melanin synthesis pathways?

Round 2

Reviewer 1 Report

My former concerns have been addressed. This study is well-performed. 

Author Response

My former concerns have been addressed. This study is well-performed.

Response: Thank you very much for your insightful comments and suggestions on our manuscript. We hope to have more opportunities to learn from you in the future.

Reviewer 3 Report

Currently, it is impossible to control the changes done by the authors in answering my requests.

The authors did not upload a clear version of the corrected manuscript.

The lane they mention does not correspond to those of the manuscript.

Furthermore, the authors mention that some part of the manuscript or figure has been completely deleted but this is not possible to appreciate in the new version they uploaded.

I would suggest that the authors will answer my request by mentioning the changes. For example, the sentence before at lane..... and now at lane .... has been changed in this way........

Otherwise, it is impossible to understand to correct potential improvement of the manuscript as mostly all the lanes do not correspond to what the authors mention in their answer.

I am sorry but in this way, I cannot accept the manuscript.

Round 3

Reviewer 3 Report

Dear Reviewer,

Thank you very much for your critical comments and thoughtful suggestions on our manuscript entitled “MITF contributes to the body color differentiation of green, purple and white sea cucumbers Apostichopus japonicus through expression differences and regulation of downstream genes” (Manuscript ID: biology-2049586). Those comments are all valuable and very helpful for revising and improving our paper, as well as the important guiding significance to our research. We have studied comments carefully and have made correction which we hope meet with approval. Revised portion are marked in the manuscript using the “TrackChanges” function (MS Word version), and the clean unmarked modified version (PDF version) is also uploaded. Below you will find our point-by-point responses to the reviewers’ comments/questions:

The manuscript of Lili Xing et al. focuses on the role of MITF as a key transcription factor involved in melanocyte development and melanin synthesis pathway in sea cucumber Apostichopus japonicous. This study is very interesting as the author used different methods to determine the potential downstream target genes regulated by MITF in mediating the different bodies colour of the sea cucumber.

However, although I would like to see this paper published in the biology journal, I must say that major revisions are needed before being accepted.

Response: We kindly thank the reviewer for the insightful comments and suggestions which is useful for us to improve the quality of our manuscript. We have revised the manuscript and addressed the issues as follows.

(1) The results sections are described in a too synthetic way and need to be better

explained also in the context of what has been found in other species.

Response: We kindly thank the reviewer for the perspicacious comments. As suggested, we have revised the results section. As this section is too long, please see Lane 237-362 in clean PDF version for details. (see Lane 237-362)

OK

2) Lane69: please rephrase this sentence

Response:Thanks for the reviewer’s comments. The sentence before at lane 68-70 and now at lane 67-70 has been changed in: “Additionally, it activates genes involved in the synthesis of melanin as well as the development and intracellular movement of melanosomes. Melanosomes is lysosome-related organelle and responsible for melanin synthesis and storage.” (see Lane 67-70).

Thank you for the corrections. However, I think that the authors should add appropriate references

(3) Lane 76: In relation to DNA fragments I would change it in: potential DNA binding sites

Response: We kindly thank the reviewer for the insightful comments. We have revised it as suggested. The “DNA fragments” before at lane 76 and now at lane 76 has been changed in “potential DNA binding sites”. (see Lane 76)

The authors performed the changes according to my suggestions.

(4) Lane 80: I think that rather than MITF candidate genes the authors should change it to potential enhancers regulated by MITF

Response: Thanks for the reviewer’s suggestions. The DNA sequences bound by MITF may include not only enhancers but also promoters and other regions (Goding, Arnheiter, 2019), so "MITF candidate genes" or "target genes" may be more consistent with research results. However, this sentence before at lane 80-81 and now at lane 80-82 has been revised for clarification as follows: “ChIP-seq profiles recently discovered a large number of candidate MITF targets across different categories, including DNA replication and repair, cell proliferation, and mitosis”. (see Lane 80-82) OK

(5) Lane 91: Again, I think that the authors should write enhancer potentially regulated by MITF rather than target genes

Response: We kindly thank the reviewer for the careful comments. The DNA sequences bound by MITF may include not only enhancers but also promoters and other regions (Goding, Arnheiter, 2019), so "MITF candidate genes" or "target genes" may be more consistent with research results. However, this sentence before at lane 90-91 and now at lane 91-92 has been revised for clarification as follows: “What’s more, ChIP-seq assay were employed to analyze genes regulated by MITF in three color morphs of sea cucumbers”. (see Lane 91-92)

OK

(6) Lane 102: It is missing the letter related to the purple A. japonicus

Response: Thanks for the reviewer’s careful comments. We have revised it. The “purple A. japonicus” before at lane 102 and now at lane 103 has been changed in “purple A. japonicus (P)”. (see Lane 103).

OK

(7) In all the methods sections very often, the authors use the ul to describe the amount of RNA; dNTP, OLIGO etc. instead of the concentration. I think that they should provide the exact concentrations used where it is possible.

Response: Thanks for the reviewer’s comments. To avoid redundancy and repetition, we have cited the reference (Xing et al., 2018). As this section is too long, please see Lane 111-123 in clean PDF version for details. (see Lane 111-123)

OK

(8) The authors should also specify which tissue they used for RNA extraction and how they performed the extraction (i.e. Tryzol, KIT). Did the author perform the primer efficiencies? Furthermore, they should also provide the primer sequence of the actin reference gene that is missing.

Response: We kindly thank the reviewer for the perspicacious comments. Total RNA of each sample was isolated from the sea cucumber body wall using RNeasy Kit (Qiagen, Texas, USA). β-actin was used as a reference gene for internal standardization. The forward primer of β-actin was 5’-CATTCAACCCTAAAGCCAACA-3’, and the reverse primer was 5’-TGGCGTGAGGAAGAGCAT-3’. And we performed the efficiencies for all primers. Since this section is a new addition, it does not have the previous lane numbers, which are now 112-113,116-119. (see Lane 112-113, 116-119)

OK

(9) Lane 150: The authors should better describe the MITF antibody used. Is the antibody produced by them? Or it has been used as an antibody by a company?

Response: We kindly thank the reviewer for the perspicacious comments. As suggested by the reviewer, we have added the detailed information in the manuscript. We commissioned Sangon Biotech Co., Ltd. (Shanghai, China) to produce MITF antibodies. The full-length sequence expressed protein of sea cucumber MITF was used as immunogen to stimulate rabbits and polyclonal antibody was obtained. This MITF antibody was specific in sea cucumber. Since this section is a new addition, it does not have the previous lane numbers, which are now 142-145. (see Lane 142-145).

Thanks for the answer and corrections. I would suggest that the authors specify if they provided a construct containing the full-length sequence of the MITF protein to the company and then expressed it before use for injection or if the authors provided a protein that they previously expressed in vitro. Please specify it in the material and methods. Moreover, I would suggest that the authors provide also in the supplementary data the full-length sequence of the MITF protein or alternatively the NCBI accession number if it has been deposited as I did not find it in the manuscript.

(10) Lane 173: The authors should provide more information on the MITF probe. How they produced it? Which kit did they use?

Response: We kindly thank the reviewer for the perspicacious comments. As suggested by the reviewer, we have provided the information in the manuscript. The probe sequence was synthesized artificially, and the FAM marker was added to the 5ʹ during the synthesis. The type of this probe was an oligonucleotide probe with higher hybridization efficiency. Since this section is a new addition, it does not have the previous lane numbers, which are now 171-173. (see Lane 171-173)

OK

(11) Lane 202: The authors should better specify the second antibody features and a more detailed description of the protocol with the exact timing of the different steps. Alternatively, they should use a reference if they used the same methods.

Response: We kindly thank the reviewer for the perspicacious comments. According to suggestion, we have added the following information to this section (highlighted in red). However, the results of immunohistochemistry were considered redundant and confusing by the reviewers and did not have much impact on the topic of the paper. After a comprehensive and serious discussion by the authors, we removed the immunohistochemistry section.

The body wall samples of white, green, and purple A. japonicus were fixed and embedded in paraffin. They were sliced using a microtome (Leica RM2125RT; Leica, Wetzlar, Germany) into 5_ _μm_ _t_h_i_c_k_ _s_e_c_t_i_o_n_s_,_ _w_h_i_c_h_ _w_e_r_e_ _p_l_a_c_e_d_ _o_n_ _s_l_i_d_e_s_,_ _d_e_w_a_x_e_d_ _(15 min each time, two times), hydrated (gradient ethanol: 100%, 95%, 85%, 75%, 5 min respectively), and washed with dH2O (5 min). The slides were immersed in trisodium citrate antigen retrieval solution (4 min). After washing, an immunohistochemical oily pen was used to circle the tissue sample position, and 3% hydrogen peroxide-methanol was added dropwise for 15 min at room temperature. The sections were washed again (5 min each time, three times) and blocked with immunostaining blocking solution (45 min). After adding the primary antibody (4°C overnight), the samples were rewarmed (room temperature for 60 min), washed (5 min each time, four times), and then HRP-labeled secondary antibody (goat anti-rabbit; Shanghai Biological Engineering Co., Ltd) w_a_s_ _a_d_d_e_d_ _d_r_o_p_w_i_s_e_._ _A_ _3_,_3_ʹ-Diaminobenzine chromogenic solution was used for color development, followed by hematoxylin redyeing. The sections were decolorized in 1% hydrochloric acid-ethanol for 10–15 s and then quickly removed. The reaction was terminated in distilled water, followed by immersion in anti-blue in phosphate buffered saline (PBS) with Tween (pH 7.4) for 5–10 min. After the sections were sealed, the slides were viewed and photographed using an Olympus BX53 microscope.

OK

(12) Lane 211-214: The authors should insert information on the specific tissue used for the ChIP assay and more detailed information on the timing and intensity of sonication as well as on the type of machine used.

Response: Thanks for the reviewer’s suggestions. The detailed information has been provided as suggested. The sentences before at lane 211-214 and now at lane 193-198 have been changed in: “First, 1 g of sea cucumber body wall tissue was cross-linked with 1% formaldehyde for 10 min at 37°C with constant shaking. Then, the addition of 125 mM glycine prevented cross-linking. Chromatins were collected on ice after samples were lysed. The collected chromatins were sonicated by Bioruptor (Diagenode, Liège, Belgium) at standard mode for 30 sec to obtain soluble sheared chromatin (average DNA length of 200–500 bp).” (see Lane 193-198).

OK

(13) Figure 1-2: The statistical analyses should be better explained. The lowercase letters are confusing because it is not easy to understand which comparison the authors are evaluating with the t-test.

Response: We kindly thank the reviewer for the perspicacious comments. As suggested by the reviewers, Figures 1 and 2 have been merged into one figure with two

panels, and the further details have been included to the manuscript. In addition, we have conducted the correlation test between mRNA and protein expressions, which showed that they were significantly correlated (P<0.01). The original Figure 1 and Figure 2 have been merged into the current Figure 1, and the results of this part before at lane 251-259 has been change in lane 238-257. (see Figure 1, Line 238-257)

OK

(14) Figure 3: The authors did not mention how negative control has been performed. Furthermore, the authors should specify which nuclear dye was used in figure W2-W3-G2-G3-P2 and P3. Moreover, they should provide a picture of the negative control without any nuclear staining as performed for the pictures W1-G1-P1. Finally, a panel with nuclear dye and fluorescent staining of the negative control (merge) should be added as they did for W3-G3-P3.

Response: We kindly thank the reviewer for the perspicacious comments. We performed negative control without fluorescent probes of MITF (panels W2, G2, and P2). The nuclear dye used in panels W2-W3-G2-G3-P2 and P3 was DAPI. The negative controls without fluorescent probes of MITF indicated the tissue structure of the sea cucumber body wall according to the DAPI nuclear dye. In addition, the negative control without any nuclear staining was black and could not provide more information, so it was not added in this study. The previous Figure 3 has been changed in the current Figure 2 and the results of this part before at lane 274-280 has been change in lane 259-268. (see Line 259-268, Figure 2)

I do not agree with the authors. Showing a picture with only DAPI is not a negative control. It is just a picture showing the nuclei in the tissue of interest. For negative control, the authors should perform an experiment of in situ hybridization with a sense probe.

(15) Figure 4: The authors should provide a picture of the negative control as without it is impossible to judge the specificity of the signal. In addition, all the images should be oriented in the same direction. P1 and P2 indeed are oriented in a different direction with respect to W1-W2; G1-G2. Finally, they should better explain which are the differences between W1 and W2 and W3 as also as the other images in the panel.

Response: We kindly thank the reviewer for the perspicacious comments. Considering the comments of the three reviewers, we deleted the content related to the

immunohistochemistry experiment from this study after serious discussion. The decision was made mainly based on the following considerations: (1) In situ hybridization had already clarified the MITF expression location, and the results of immunohistochemistry were “quite confusing and redundant”; (2) there was no difference in the MITF expression location among the three color types of sea cucumbers, and the immunohistochemistry experiment could not provide more information; (3) there was a lack of negative control in the results of immunohistochemistry; (4) deleting the immunohistochemistry results did not affect the theme and conclusion in this study, but could maintain the rigor and accuracy of the manuscript.

(16) Lane 320: The author should specify what the KEGG acronym means as they do not sentence it before.

Response: We kindly thank the editor for the careful comments. The KEGG acronym was already specified in Line 227. (see Line 227)

I apologize the authors are right.

(17) Figure 5-6: The authors should completely prepare again the pictures in a more readable way. It is almost impossible to read them.

Response: We totally agree with the reviewer’s comments. As suggested by the reviewer, we have revised these two figures. After modification, the previous Figure 5 and Figure 6 are now Figure 3 and Figure 4 respectively. (see Figure 3, 4)

Ok

(18) Table 4-5: These tables are too magnified and need to be rearranged.

Response: Thanks for the reviewer’s comments. We have modified Table 4 and Table 5, and only the top 5 known motifs were displayed in Table 4 and Table 5, while the rest HOMER enrichment results were moved to the Supplementary materials. (see Table 4, 5, S7, S8)

OK

(18) Finally, the manuscript will benefit from some validation of the most relevant identified enhancer. It will be interesting for example to perform some

Electrophoresis Mobility Shift Assay (EMSA) to demonstrate the ability of the Apostichopus japonicus MITF protein to bind the identified potential binding site.

Response: We respectfully agree with the reviewer’s comments. We cherish this opportunity very much and have tried our best to improve our manuscript. All the authors have seriously discussed about this suggestion. We understand that the addition on EMSA experiments would make this study more attractive. However, the current research base of sea cucumbers is relatively weak compared to other model organisms, and supplementing EMSA experiments requires continuous figuring out, a process that may take a long time. In future studies, we will conduct validation experiments such as EMSA to improve the research.

Although, I do not fully agree on the difficulties I can understand that the authors decided to do not perform these experiments for a further validation.

Thank you again for your suggestions, and hope to learn more from you.

References

Goding, C.R., Arnheiter, H., 2019. MITF-the first 25 years. Genes & Development. 33, 983-1007.

Xing, L.L., Sun, L.N., Liu, S.L., Wan, Z.X., Li, X.N., Miao, T., Zhang, L.B., Bai, Y.C., Yang, H.S., 2018. Growth, histology, ultrastructure and expression of MITF and astacin in the pigmentation stages of green, white and purple morphs of the sea cucumber, Apostichopus japonicus. Aquaculture Research. 49, 177-
